# 1,2,4-Thiadiazolidin-3,5-Diones as Inhibitors of Cysteine Proteases

**DOI:** 10.3390/molecules30193896

**Published:** 2025-09-26

**Authors:** Maria Aparecida Juliano, Marco Persico, Beatrice Severino, Giuseppe Tumbarello, Debora Okamoto, Karolina Rosa Fernandes, Gabriel Trigo, Aparecida Sadae Tanaka, José Thalles Lacerda, Oleh Tkachuck, Angela Corvino, Ferdinando Fiorino, Antonia Scognamiglio, Francesco Frecentese, Vincenzo Santagada, Stefania Vertuccio, Giuseppe Caliendo, Luiz Juliano, Caterina Fattorusso

**Affiliations:** 1Department of Biophysics, Escola Paulista Medicina, Universidade Federal São Paulo, Rua Tres de Maio, 100, São Paulo 04044-020, SP, Brazil; ma.juliano@unifesp.br (M.A.J.); karolina.rosa@unifesp.br (K.R.F.); gabriel.trigo@unifesp.br (G.T.); thalles_lacerda2@hotmail.com (J.T.L.); 2Department of Pharmacy, School of Medicine, University of Naples «Federico II», Via D. Montesano, 49, 80131 Napoli, Italy; marco.persico@unina.it (M.P.); bseverin@unina.it (B.S.); giuseppe.tumbarello@unina.it (G.T.); oleh.tkachuck@unina.it (O.T.); angela.corvino@unina.it (A.C.); fefiorin@unina.it (F.F.); antonia.scognamiglio@unina.it (A.S.); francesco.frecentese@unina.it (F.F.); santagad@unina.it (V.S.); stefania.vertuccio@unina.it (S.V.); caliendo@unina.it (G.C.); 3Department of Pharmaceutical Sciences, Universidade Federal de São Paulo, Rua São Nicolau, 210, Diadema 09913-030, SP, Brazil; debora.okamoto@unifesp.br; 4Department of Biochemistry, Escola Paulista Medicina, Universidade Federal São Paulo, Rua Tres de Maio, 100, São Paulo 04044-020, SP, Brazil; astanaka10@unifesp.br; 5Department of Pharmaceutical Sciences, Federal University of Pernambuco, Rua Artur de Sá, Recife 50740-521, PE, Brazil

**Keywords:** 1,2,4-thiadiazolidin-3,5-diones, cysteine proteases inhibitors, 3CLpro, PLpro, SARS-CoV-2, Cathepsin L, Papain, molecular modeling

## Abstract

A focused library of 1,2,4-thiadiazolidin-3,5-diones (**THIA-1**–**10**), previously characterized as hydrogen sulfide (H_2_S) donors, was evaluated for inhibitory activity against cysteine proteases. We included two key cysteine proteases aiming at antiviral drug development—SARS-CoV-2 3CLpro (Mpro) and PLpro—alongside reference enzymes Papain and Cathepsin L. The compounds exhibited distinct selectivity profiles and inhibition mechanisms. The ability to act as covalent inhibitors of 3CLpro in the nanomolar range is of particular interest, with compounds **THIA-6**, **-7**, and **-10** proving to be the most potent inhibitors of the series, and compounds **THIA-1**, **-2**, and **-8** proving to be the most selective with respect to the other proteases. We explored the molecular bases of the observed activity profile of **THIA-1**–**10** through computational studies, which supported and complemented the experimental findings, paving the way for future structure optimization. The results highlight that inhibitory potency depends not only on electrophilicity but also on the ability to access the catalytic cysteine within the active site. The dual functionality of **THIA-1**–**10** as H_2_S donors and selective cysteine protease inhibitors underscores its potential as a promising lead for therapeutic development.

## 1. Introduction

Cysteine proteases have emerged as critical therapeutic targets, particularly in the context of viral infections such as SARS-CoV-2. These enzymes play essential roles in viral replication and immune evasion, making them attractive candidates for the development of selective covalent inhibitors. 1,2,4-Thiadiazolidin-3,5-diones (THIAs) are a versatile class of heterocycles with diverse pharmacological applications. They have been reported as inhibitors of glycogen synthase kinase 3β (GSK3β) [1], regulators of G protein signaling (RGS) proteins [2], and hydrogen sulfide (H_2_S) donors [3]. The underlying mechanisms of these activities are based on the reaction of the sulfur atom of THIAs with free sulfhydryl groups (SH) contained in compounds that release H_2_S [3] or with the catalytic cysteine residue in enzymatic reactions [2]. Given this reactivity, inhibition of cysteine proteases is an expected activity for THIA derivatives. However, 4-(4-fluorobenzyl)-2-p-tolyl-1,2,4-thiadiazolidine-3,5-dione (CCG-50014) failed to inhibit Papain [4]. This intriguing observation led us to consider that the covalent inhibition requires, besides the reaction of the THIA ring with the catalytic cysteine, the adequate fitting of the compound in the active site, as defined for Papain by Schechter and Berger [5] and mapped in other cysteine proteases with libraries and systematic series of synthetic peptide substrates [6,7,8,9,10].

Based on these considerations, we selected a focused library of ten THIA derivatives (**THIA-1**–**10**) (Table 1), previously characterized as H_2_S donors [3], to evaluate their inhibitory activity against cysteine proteases with distinct substrate specificities. We chose two reference enzymes—Papain and Cathepsin L—and two SARS-CoV-2 proteases: the Chymotrypsin-like protease (3CLpro or Mpro), essential for viral polyprotein processing, and the papain-like protease (PLpro), which exhibits deubiquitinating and deISGylating activities. The aim of this study is twofold: (i) to assess the potential of **THIA-1**–**10** as selective covalent inhibitors of cysteine proteases and (ii) to elucidate the molecular basis of their activity through computational approaches, thereby providing a foundation for future structure-based optimization.

Papain and Cathepsin L share 41% sequence identity and exhibit similar crystal structures [11,12], with well-characterized S1 to S1’ subsites that accommodate substrate residues and contribute to binding specificity [13,14,15]. 3CLpro is active only in its dimeric form [16]. Each homodimer comprises three domains: domains I and II consist of antiparallel β-barrels that form the chymotrypsin-like fold, while domain III facilitates dimerization. The catalytic center consists of a Cys^145^–His^41^ dyad. Crystal structures of 3CLpro–inhibitor complexes reveal that the hydrophobic residues from P_3_ to P_1′_ of the inhibitors fit into the corresponding protease subsites S_3_ to S_1_^’^ [17,18,19,20,21]. PLpro is a cysteine protease with a canonical catalytic triad composed of Cys^111^, His^272^, and Asp^289^. Its S_2_ and S_1_ subsites are located within a narrow channel that accommodates the Gly–Gly motif, enabling the enzyme to perform deubiquitinating and deISGylating activities [22,23,24,25,26,27].

We initially evaluated the stability of **THIA-1**–**10** in organic solvents and buffer solutions. Subsequently, we determined the inhibition type for each protease using **THIA-3** as a reference compound, followed by the calculation of IC_50_ values for all other THIA derivatives. Despite the limited number of compounds investigated in the present work, they exhibited notable in vitro potency and selectivity toward therapeutically relevant cysteine proteases. Accordingly, computational studies—including conformational analysis, density functional theory (DFT) calculations, molecular docking simulations, and structural and bioinformatics analyses—were performed to investigate the molecular bases of the observed activity, laying the groundwork for future optimization of the identified hits.

## 2. Results and Discussion

### 2.1. Stability of THIA Derivatives

We chose **THIA-3** as a reference due to its simplicity in larger-scale synthesis to evaluate the stability of this class of compounds. In DMSO with 1% water, **THIA-3** decomposed to 1,3-diphenylurea, a surprising result that occurred surprisingly quickly (Appendix A). However, in acetonitrile (ACN), **THIA-3** remained stable even when diluted in water or phosphate buffer. In the Tris buffer, decomposition to 1,3-diphenylurea was very slow, with some decomposition only apparent after 12 h of incubation (Appendix A). We also evaluated the stability of 10 mM stock solutions of the ten THIAs in DMSO stored in a freezer (−20 °C) by Liquid Chromatography Mass Spectrometry (LC-MS). **THIA-1**, **THIA-2**, **THIA-7**, **THIA-8**, and **THIA-10** were the most resistant, indicating that the stability of the THIA derivatives in this condition depends on R or R’ substituents (Appendix A). DMSO is also considered a reagent [28], but we have not explored the decomposition of THIA compounds in DMSO further, as it falls outside the scope of this work. These results justified using ACN as the organic solvent of the THIA derivatives and 100 mM phosphate as the buffer in all experiments with the cysteine proteases.

### 2.2. Inhibitory Activities of THIAs on Cysteine Proteases

#### 2.2.1. Selection of Substrates

Z-Phe-Arg-MCA is a commercially available fluorescent substrate for Papain and Cathepsin L. Both proteases hydrolyze it at the Arg-MCA peptide bond with high k_cat_ and K_m_ values. 3CLpro and PLpro cleave the SARS-CoV-2 polyprotein into sixteen mature components, termed nonstructural proteins (NSPs). We synthesized the fluorescence resonance energy transfer (FRET) peptide Abz-SAVLQSGFRK(Dnp)NH_2_ [Abz = ortho-aminobenzoic acid; K(Dnp) = N-ε-(2,4-dinitrophenyl) Lys]) based on the SARS-CoV-2 3CLpro optimum substrate sequence, as we previously reported [16]. For PLpro, we initially synthesized and assayed a series of FRET peptides of general structure Abz-peptidyl-Q-EDDnp (EDDnp = ethylene diamine 2,4-dinitrophenyl) [29] with sequences of SARS-CoV-2 polyprotein containing the sites nps2/3 and nps3/4 cleaved by PLpro and four peptides containing the C- terminal sequence of two ubiquitins linked to εNH_2_-caproic acid carrying Q-EDDnp or the peptide sequence of two ubiquitinated protein (Appendix A). Based on these results, we selected Abz-TLKGGAPIK-Q-EDDnp as the substrate for PLpro.

#### 2.2.2. Kinetic Mechanism of Inhibition

Initially, we identified the type of inhibition in each protease using **THIA-3** as a reference compound for all the tested THIA derivatives. The Papain activity decreased with progressive **THIA-3** pre-incubation time (Appendix A), indicating irreversible inhibition that also occurred with Cathepsin L. Using a previously reported protocol [30], we observed a covalent irreversible inhibition of **THIA-3** on Papain and 3CLpro, reverted by reducing agents (Appendix A), and reversible inhibition of PLpro (Appendix A). Due to the susceptibility of the THIA derivatives to reducing agents, such as Cys or DTT (Appendix A), the proteases Papain, Cathepsin L, 3CLpro, and PLpro were separately pre-activated by 5 mM DTT during 30 min, which was removed by gel filtration, forming stock solutions of the activated proteases, and then kept at −20 °C under an N_2_ atmosphere for future use.

Figure 1 shows the reactions of **THIA-3** and **THIA-10** with Papain, as well as **THIA-3** with 3CLpro, which are compatible with a two-step covalent enzyme inactivation mechanism. This is confirmed by kinetic analyses of the reaction of **THIA-3** with Papain, used as a diagnostic method as previously reported [31]. The kinetic constant values of the reaction of the ten THIA compounds with Papain, for the reaction paths shown below, are reported in Appendix A.

These results confirmed a two-step, irreversible bimolecular reaction, namely, inhibitor binding and the formation of a covalent inhibitory complex, as shown in Figure 1A. The assistance in this reaction of the imidazole group of His that belongs to the cysteine protease cysteine catalytic triad is shown in Figure 1B.

The nucleophilic attack of the catalytic cysteine to the S_1_ sulfur atom of the THIA ring and the subsequent adduct formation is superimposable to what was previously proposed by us for the reaction with free L-cysteine in solution [3] (Figure 1). However, the reaction of the resulting adducts with a second cysteine observed in the solution of this free amino acid, resulting in the release of H_2_S, is not conceivable with the cysteine protease.

To compare the inhibition efficiency of the THIA derivatives on the four cysteine proteases, we fixed the pre-incubation time at 10 min to obtain the IC_50_ values. Table 2 shows the obtained results.

### 2.3. IC_50_ Values with Papain and Cathepsin L

The inhibitory activity trend of **THIA-1**–**10** is very parallel for papain and cathepsin L, even though most of the compounds were more active on papain, as reported in Table 2. In particular, **THIA-1**, **THIA-2**, and **THIA-8** were the worst inhibitors on both proteases, with IC_50_ in the μM range, while all other THIAs showed IC_50_ values in the nM range. The common feature that distinguishes **THIA-1**, **THIA-2**, and **THIA-8** from the other tested compounds is the presence of an alkyl group as an R substituent. In contrast, in the other compounds, R is a phenyl ring (with or without para-substitution). In the papain family, the S_1_ subsite is strictly conserved (Appendix A), showing a preference for positively charged residues (i.e., arginine and lysine). On the contrary, the S_2_ subsite is quite different in shape and size and represents the predominant factor in defining substrate or inhibitor specificities [7,11,12,14,32] (for more details, see the paragraph “Investigation of the molecular bases of THIA selectivity”). Finally, papain and cathepsin L also present some differences in the S_3_ subsite (Appendix A) due to their broader interaction specificity [8]. **THIA-1** and **THIA-2**, with a benzyl group and **THIA-8**, with a cyclohexyl group, as R substituents, proved to be the least effective inhibitors on both enzymes, suggesting that the effect exerted by the presence of an alkyl carbon at N4 on the reactivity of the THIA ring may be involved in the drop in inhibitory activity. Interestingly, compound CCG-50014, which exhibits a complete absence of papain inhibition [4], has a benzyl group as an R substituent, similar to **THIA-1** and **THIA-2**, but with fluorine in the para position. Possibly, fluorine in the para position of the benzyl group further alters its properties [33,34,35], impairing the CCG-50014 reaction with papain. In contrast, **THIA-7** with the 4-chlorophenyl group as the R substituent inhibited papain and cathepsin L with IC_50_ values in the nanomolar range, similar to the unsubstituted analog **THIA-4**. The halogen at the para position of the phenyl ring did not impair the inhibition processes, indicating that the drop in the activity is related to the presence of the benzyl group.

**THIA-6**, bearing 4-methoxyphenyl substituents at both N2 and N4, is one of the most effective inhibitors of Papain. Notably, its IC_50_ (0.01 μM) is lower than that of **THIA-4** (0.07 μM), containing just one 4-methoxyphenyl at N2 (R’ substituent), which is, in turn, lower than that of **THIA-3** (0.14 μM), presenting two unsubstituted phenyl rings as substituents. These results suggest that both 4-methoxyphenyl groups of **THIA-6**, at R and R’ positions, may have favorable interactions with Papain.

Finally, **THIA-10** is another of the best inhibitors of Papain. Its IC_50_ (0.01 μM) is lower than that of **THIA-3** (0.14 μM), and they differ only by the presence of fluorine in the para position of the phenyl group at the R’ substituent. The electron-negative inductive effect of fluorine in the para position of the phenyl group in the R’ substituent is supposed to speed up the nucleophilic attack of the papain catalytic Cys on the sulfur atom (S1) of the THIA ring. This electronic effect of the R’ substituent was evident in the release of H_2_S by reaction of **THIA-10** with Cys [3]. On the other hand, the inhibition of Cathepsin L by **THIA-10** yielded an IC_50_ similar to that obtained with **THIA-3**, while the IC_50_ values of **THIA-1**, **-2**, and **-8** are ten times higher than those obtained with Papain.

To investigate the role of S_2_ subsite fitting in the selective inhibition of cysteine proteases by THIA derivatives, we evaluated their activity against bromelain. Indeed, this protease is unique among cysteine proteases, as it requires substrates containing a positively charged residue at the P_2_ position due to the presence of Glu^68^ in its S_2_ subsite [36]. Notably, we observed inhibition with the pre-incubation during 10 min of 10 μM only with **THIA-7** (IC_50_ = 2.0 ± 0.1μM), which contains the *para*-Cl-phenyl group in the R position. The Cl at the para position induces dipole charge resonance structures in the phenyl group, establishing a favorable interaction with the Glu^68^ carboxylate group in the S_2_ subsite (Appendix A; and see Section 2.5 “Molecular Modeling Studies”).

THIA compounds seem to have comparable reactivity as the classical known irreversible cysteine protease inhibitors, such as L-trans-Epoxysuccinyl-leucylamido (4-guanidino)butane (E-64) and its analogues, which are also more effective irreversible inhibitors of papain than cathepsin L [37], similar to what we observed with the THIA compounds. The poor inhibition of Bromelain by E-64 [36], due to the presence of the negative charge of Glu^68^ at the bottom of S_2_ subsite, was also observed with THIA compounds since bromelain was inhibited only by **THIA-7**, possibly due to the Cl substitution in the para position of the phenyl group at N4 of **THIA-7**, which allows better fitting to the S_2_ subsite. Another relevant comparison is with the peptidyl diazomethane Z-Phe-Phe-CHN2, which is a better inhibitor of cathepsin L than Z-Phe-Ala-CHN2 [37], because Z-Phe-Phe-CHN2 occupies both the S_1_ and S_2_ subsites of cathepsin L. A similar situation is observed with **THIA-3** (IC_50_ = 0.12µM), which has two aromatic substituents, in contrast to **THIA-8** (IC_50_ = 27 µM), which has an aromatic substituent only in N2 of the THIA ring.

### 2.4. IC_50_ Values with SARS-CoV-2 Cysteine Proteases

The IC_50_ inhibition values of 3CLpro with the ten THIA derivatives (Table 2) are in the nanomolar (nM) range, demonstrating remarkable performance as protease inhibitors. **THIA-6**, **THIA-7**, and **THIA-10** are the best inhibitors, with IC_50_ values ten times lower than those of the other assayed THIAs, except for **THIA-5**. In contrast with Papain and Cathepsin L, the proteases 3CLpro and PLpro have, respectively, 12 and 11 Cys residues (Figure 2). The room-temperature crystal structure of 3CLpro demonstrated that only catalytic Cys^145^ is oxidized. At the same time, the other surface Cys remains reduced, and only Cys^145^ and Cys^156^ react with the alkylating agent N-ethylmaleimide [20]; then, the irreversible inhibition of 3CLpro by THIA derivatives must be due to their reaction with Cys^145^ in the catalytic site [21].

To correlate the 3CLpro substrate specificity with its inhibition by the THIA derivatives, we synthesized and assayed a library of FRET peptides with 3CLpro, using Abz-SAVQSSGFRK(Dnp)-NH_2_ as the reference sequence that spans the S_4_ to S_3_’ subsites. The amino acid sequence of this peptide spans the cleavage between the nonstructural protein 4 (nsp4) and the N-terminal side of 3CLpro (SAVLQ^3240^↓S^3241^GFRK), which is highly susceptible to 3CLpro [16]. Figure 3 shows the hydrolytic activity of 3CLpro on each peptide.

At position P_1_, 3CLpro hydrolyzed only peptides containing Gln and His, with the latter being a very poor substrate. The carboxamide nitrogen of Gln is isosterically equivalent to Nτ-imidazole; in contrast, the nitrogen of Asn is isosteric to Nπ-imidazole of His (Figure 2), but the peptide with Asn is resistant to 3CLpro. Notably, there is a very restrictive specificity to the carboxamide group of Gln of the S_1_ subsite of 3CLpro.

The S_2_ subsite of 3CLpro is also highly selective, efficiently cleaving only the peptide with Leu. Little hydrolysis occurred with the hydrophobic amino acids Met and Phe, and complete resistance to hydrolysis was observed with all other amino acids. The S_3_ subsite is more tolerant and accepts almost all classes of amino acids, with the highest hydrolysis with Val, Leu, and Arg, while S_4_ significantly prefers Ala, suggesting it is a shallow subsite. The subsites S_1_’ and S_2_’ exhibit selectivity for amino acids with small side chains, with a preference for Gly, and finally, S_3_’ accepts any amino acid. We also synthesized and assayed a second library containing His at the P1 position with 3CLpro (Appendix A), and the profile of specificity was very similar to that of Gln in the P1 position, except that peptides with Met, Phe, and Thr at the P2 position were more efficiently hydrolyzed. The THIA derivatives are short compared to the FRET peptides; still, as covalent inhibitors, the THIA ring must occupy the S_1_ subsite for the nucleophilic attack by Cys^145^ and, due to the hydrophobicity of the S_2_ subsite, it very probably binds one of the THIA substituents (R or R’).

While we were developing this work, a detailed analysis of the specificity of the S_6_-S_6_’ subsites of 3CLpro using a comprehensive methodology called Proteomic Identification of Cleavage site Specificity (PICS) was published [38]. Overall, our results align with those obtained by PICS, and the substrate Abz-SAVLQSGFRK(Dnp)-NH_2_ used for 3CLpro assays proved adequate.

The IC_50_ values of THIA derivatives with PLpro are all in the μM range, with about half of them exceeding 10 μM. (Table 2). The observed reversible inhibition (Appendix A) is due to the inaccessibility of subsites S_1_ and S_2_ to larger molecular structures than glycyl-glycine, resulting in weaker inhibition of THIAs compared to the covalent inhibitors that resulted from the nucleophilic reaction with Cys^111^ in the catalytic site [39]. This reversible inhibition also indicates that the other ten Cys residues of PLpro, as shown in Figure 2, are not reactive toward THIA derivatives under the conditions we tested. The noncovalent deubiquitinase inhibitor 5-amino-2-methyl-N-[(1R)-1-(1-naphthalenyl)ethyl]benzamide (GRL0617) inhibited PLpro (IC_50_ = 0.6 μM) through an unusual mechanism, binding to the S_4_–S_3_ subsites and inducing a conformational change in the loop that closes the access to the catalytic site [23]. The **THIA-5** and **THIA-8**, with lower IC_50_ values (3.9 μM and 4.2 μM, respectively), contain some components, such as GRL0617. The good structural overlap between GRL0617 and **THIA-5** in complex with PLpro supports the putative binding of both compounds to the same allosteric site (for more details, see the paragraph “Investigation of the molecular bases of THIA selectivity”; Appendix A).

### 2.5. Molecular Modeling Studies

To elucidate the molecular basis of the inhibitory activity of THIA derivatives and to gain critical insights for future structural optimization, a comprehensive set of computational studies was conducted. Initially, the conformational and electronic properties of the compounds listed in Table 1 were evaluated using a combination of Molecular Mechanics (MM) and Density Functional Theory (DFT) calculations. Subsequently, molecular docking simulations with 3CLpro were performed on a representative subset of derivatives, ranging from the most active (**THIA-7**) to the least active (**THIA-8**). These initial docking studies were designed to simulate the adaptation of the inhibitor within the enzyme’s active site, with the specific aim of positioning the reactive warhead in proximity to the catalytic cysteine, thereby facilitating the nucleophilic attack required for covalent bond formation. To rationalize the observed selectivity with respect to other cysteine proteases, a bioinformatic and structural analysis was conducted. Finally, covalent docking techniques were employed to explore the potential adaptation of the inhibitor within the enzyme’s active site following the formation of a covalent bond.

#### 2.5.1. Ligand Analysis

A stochastic conformational search procedure sampled the conformational space of THIA derivatives [40], followed by DFT full optimization of the MM conformers using the conductor-like polarizable continuum model (C-PCM) to mimic an aqueous environment [41]. The comparison of the starting MM conformers with the resulting DFT ones highlights significant structural changes, and different MM structures converge towards the same DFT conformer (Appendix A). These outcomes underline the need to use DFT methods to model highly conjugated structures, such as THIA derivatives.

In particular, the planes of the (substituted)-phenyl rings at N2 (τ1) and N4 (τ2) always assume a perpendicular orientation with respect to that of the 1,2,4-thiadiazolidin-3,5-dione ring (τ1 and τ2 ~90°). These observations indicate that the conjugation of N2 and N4 with the adjacent carbonyl groups prevails over the conjugation with the phenyl substituents, as confirmed by the nitrogen-carbonyl carbon bond length and order (partial double) compared to those of the nitrogen-phenyl carbon bond (single). Accordingly, the most negative values of the molecular electrostatic potentials are placed on the oxygen atoms of the carbonyl functions, particularly on O3 (as shown in Figure 4 and Figure 5).

According to the results of our conformational analysis, compounds with two rotatable bonds, such as **THIA-3**, **THIA-9**, and **THIA-10**, show just one conformer (Appendix A; Figure 4A), while compounds with three rotatable bonds, such as **THIA-1**, **THIA-4**, **THIA-5**, **THIA-7**, present a couple of specular, not superimposable, and isoenergetic conformers (i.e., conformational enantiomers; namely A and B in Figure 4B; Appendix A). Finally, compounds with four rotatable bonds (**THIA-2** and **THIA-6**) together with **THIA-8**, whose cyclohexyl ring can adopt two almost isoenergetic orientations, show two couples of conformational enantiomers (namely, 1A-B and 2A-B; τ2 = 180° or 0°, respectively) (Figure 5; Appendix A).

The limited energetically accessible conformational space of our THIA derivatives, as well as the presence of conformational enantiomers, are likely to play a role in their ability to fit into the cysteine protease active site. According to the frontier orbital theory, the energy of the lowest unoccupied molecular orbital (LUMO) accounts for the propensity of a molecule to acquire electrons. Hence, in order to evaluate the tendency of THIA compounds to undergo nucleophilic attack by the sulfur atom of the catalytic cysteine [42], the LUMO energy of the DFT conformers was calculated (Table 3).

The results evidence that the propensity to acquire electrons decreases (LUMO energy increases ~ 3 kcal/mol) in compounds presenting a benzyl (**THIA-1** and **THIA-2**) or a cyclohexyl (**THIA-8**) ring at N4 compared to those presenting a (substituted)-phenyl ring (**THIA-3**, **THIA-4**, **THIA-5**, **THIA-6**, **THIA-7**, **THIA-9**, and **THIA-10**). Moreover, among these latter, the value of the LUMO energy varies (~1 kcal/mol) depending on the nature of the *para*-substituents on the two phenyl rings. Electron-withdrawing substituents (i.e., *p*-Cl (**THIA-7**) and *p*-F (**THIA-10**)) decrease LUMO energy, while electron-donating substituents (such as *p*-OMe or *p*-Me; **THIA-4**, **THIA-5**, **THIA-6**, and **THIA-9**) determine an increase in LUMO energy (Table 3). By comparing the data in Table 3 with those in Table 2, it can be observed that the ability of **THIA-1**, **THIA-2**, and **THIA-8** to act as covalent inhibitors is notably lower than that of the other THIA derivatives in the case of Papain and Cathepsin L. At the same time, the inhibitory activity toward our target 3CLpro is scarcely affected. These observations suggest that, besides compound reactivity, the role of the R and R’ substituents in the ability of the compound to interact with the 3CLpro active site and correctly orient the reaction partner with respect to the catalytic cysteine plays a crucial role in the inhibitory activity toward this enzyme (see next paragraph).

#### 2.5.2. Docking Calculations

Docking studies were conducted to simulate the accommodation of THIA derivatives within the 3CLpro active site immediately prior to nucleophilic attack by the catalytic cysteine residue. Calculations were performed taking into account the activation of Cys^145^ and the interaction with His^41^, and the results were filtered based on the orientation of the reacting atoms (Figure 1). To achieve this aim, we applied a shape-based grid docking procedure (LigandFit), combined with MM energy minimization (CFF forcefield; Dassault Systèmes BIOVIA, San Diego, 2023) [43].

In order to consider all possible ligand conformers and, at the same time, preserve the DFT-calculated ligand conformation during docking (force-field-based) simulations, all conformers of **THIA-2** (four), **THIA-4** (two), **THIA-7** (two), **THIA-8** (four), and **THIA-10** (one) were used to generate separate ligand-protein starting complexes for a total of thirteen docking runs, during which the ligand conformation was tethered.

A structural and bioinformatic analysis guided the definition of the area of the protein to be explored during docking simulations (namely, the ligand binding domain area). The tertiary structure of 3Clpro consists of three domains: Domain I and II, connected by a long loop region (named LH-Loop) to Domain III (Appendix A). The substrate-binding site is present in a cleft between Domains I and II and buries the catalytic dyad (i.e., Cys^145^ and His^41^). Upon reaction of the sulfur atom of Cys^145^, the hydrogen is transferred as a proton to the nitrogen atom of the scissile peptide bond via protonation of His^41^; this latter, along with the backbone amides of Gly^143^, Ser^144^, and Cys^145^, form the so-called oxyanion hole, which assists the stabilization of the thiohemiketal intermediate [44]. The catalytic site can be divided into subsites, namely, S_1_-S_5_ and S_1_’-S^2^’ [45,46] (Appendix A); accordingly, to fully cover these subsites, the ligand binding domain area was defined by a binding site sphere of 12 Å around the mass center of the co-crystalized inhibitor of the 3CLpro-GRL-2420 complex (PDB ID: 7JKV) (Volume: 522.5 Å^3^; Appendix A). Moreover, in order to preserve the activation of the catalytic cysteine during docking simulations, we applied a tethering restraint to the hydrogen bond between Cys^145^ and His^41^.

Each of the thirteen docking runs generated up to 20 complexes (Appendix A), which were subjected to energy minimization and subsequently filtered based on the maximum distance between the ligand and enzyme reactive atoms required for nucleophilic attack (i.e., the distance between the sulfur atom S1 of the THIA derivative and the sulfur atom of Cys145 ≤ 4 Å) [47,48]. Selection criteria also included the activation of Cys^145^ by His^41^. The ligand–protein interaction energy of the selected complexes was calculated, and the resulting structures were ranked and analyzed.

Analysis of the docked **THIA-2**/3CLpro complexes shows that the ligands could approach the catalytic Cys^145^ by placing the R and R’ substituents in different subsites, resulting in four possible binding modes (named BM-I, BM-II, BM-III, and BM-IV; Figure 6, Appendix A).

However, the reaction mechanism of THIA derivatives with thiols involves cleavage of the S1–N2 bond, accompanied by protonation of N2 [2] (Figure 1), analogous to the protonation of the scissile peptide bond nitrogen [42]. As previously described, in the 3CLpro active site, this proton transfer occurs in the oxyanion hole, where a proton shifts from Cys^145^ to the substrate nitrogen via His^41^. Accordingly, for THIA N2 to be protonated, it must be positioned within the oxyanion hole—a spatial arrangement achievable only in binding mode BM-I, where the N2 and N4 substituents occupy the S_2_ and S_3_ subsites, respectively (Figure 6A). Based on this rationale, the BM-I complex with the most favorable ligand–protein binding energy was identified as the best-docked structure (Table 4 and Appendix A; Figure 7 and Figure 8 and Appendix A).

The contribution of each protein residue to the binding energy was assessed (Appendix A), and the number and types of ligand-protein interactions were also estimated using a geometry-based method [49]. It is worth noting that all THIA derivatives interact with several residues also involved in the interaction with the substrate, such as Gln^189^, His^41^, Met^49^, His^163^, His^164^, Met^165^, and Glu^166^ (Figure 7 and Figure 8 vs. Appendix A), with the strongest contribution to the binding energy (~−4 kcal/mol) being represented by the hydrogen bond between O3 and the side chain of Gln^189^ (S_2_ subsite; Figure 7 and Figure 8; Appendix A). Going into details, the OMe group at R’ of **THIA-2**, **THIA-4**, **THIA-7**, and **THIA-8** always presents the same conformation (corresponding to conformational enantiomer B) and contributes to the ligand-binding energy by establishing favourable interaction with the S_2_ subsite (particularly with Asp^187^, Met^49^, and Arg^188^; Figure 7B and Figure 8; Appendix A). The other carbonyl group (C5=O5) points toward S_1_, partially occupying and interacting (His^163^) with this subsite, more (**THIA-2**) or less (**THIA-10**), depending on the considered THIA compound (Figure 7 and Figure 8). In particular, **THIA-2**, thanks to the presence of the flexible methylene linker, shows the best accommodation with the catalytic site (best binding energy), establishing a hydrogen bond between O5 and the side chain of His^163^ (Figure 7B), thus, mimicking the interaction of the side chain of the P1 residue (Gln) of the substrate (Appendix A).

**THIA-2** and **THIA-8** show, indeed, the most favourable ligand-protein binding energies (Table 4). Interestingly, **THIA-1**, **THIA-2**, and **THIA-8**, characterized by the presence of a sp^3^ carbon attached to N4, present the highest LUMO energy among the studied THIAs, that is, the lower propensity to acquire electrons (Table 3). Our docking results suggest that their ability to fit into the 3CLpro active site by establishing strong interactions with the protein could “compensate” for their relatively lower reactivity (Figure 7B and Figure 8C; Appendix A). Indeed, **THIA-2** and **THIA-8** (as well as **THIA-1**) remain 3CLpro covalent inhibitors at submicromolar concentrations, while retaining potency against the other proteases. Hence, the presence of a sp^3^ carbon at N4 of the THIA ring decreases electron affinity but increases the fitting of the R substituent into 3CLpro S_3_ subsite, resulting in compounds (**THIA-1**, **THIA-2**, and **THIA-8**) with comparable 3CLpro inhibitory potency with respect to their phenyl counterparts (**THIA-3** and **THIA-4**) but showing improved selectivity.

On the other hand, the **THIA-10**/3CLpro complex exhibits the lowest ligand–protein binding energy among all THIA derivatives (still indicating a favorable interaction; Table 4). Interestingly, while all other THIA compounds also adopt “unproductive” binding modes (BM-II to BM-IV)—often with more favorable interaction energies than BM-I (Appendix A)—**THIA-10** is restricted to BM-I only (Appendix A). This exclusive adoption of the productive binding mode increases the likelihood of a successful interaction with the protein and may contribute to entropic advantages. In addition, the **THIA-10**/3CLpro complex displays the shortest distance between the S1 sulfur atom of the ligand and the sulfur atom of the catalytic residue Cys^145^ (Table 4). This proximity is likely favored by an interaction between the fluorine atom and the backbone of Asp^187^ and Arg^188^, which induces a shift in the ligand toward the S_2_ subsite, bringing the reactive centers closer together compared to other docked derivatives. Finally, the electron-withdrawing effect of the para-fluorine substituent, which increases the electrophilicity of the THIA core and facilitates nucleophilic attack (Table 3 and Table 4), may further contribute to the high inhibitory potency of this compound toward 3CLpro (IC_50_ = 0.02 μM; Table 2).

Thus, the ability of the studied THIAs to act as 3CLpro covalent inhibitors seems to be the result of a balance between chemical reactivity and the ability to interact with the catalytic site properly, orienting the reacting partners. In line with this hypothesis, **THIA-7**, presenting the best compromise between the calculated LUMO energy and the ligand-protein interaction energy (Table 4), resulted in the most potent 3CLpro inhibitor among the tested THIAs (IC_50_ = 0.01 μM; Table 2). On the other hand, **THIA-4**, with relatively low LUMO energy (lower than **THIA-2** and **THIA-8**, higher than **THIA-7** and **THIA-10**) and a ligand-protein binding energy value lower than **THIA-10** but higher than the other THIAs, showed an inhibitory potency against 3CLpro (IC_50_ = 0.13 μM), comparable to that of **THIA-2** and **THIA-8**. The introduction of a para-halogen substituted phenyl ring at either N4 (R; S_3_ subsite; **THIA-7**) or N2 (R’; S_2_ subsite; **THIA-10**) significantly increases electron affinity (Table 3), speeding up the reaction with the catalytic cysteine. Accordingly, **THIA-7** and **THIA-10** exhibit increased potency as 3CLpro covalent inhibitors but decreased selectivity, being also very potent against Papain and Cathepsin L (see the following paragraph).

The positive effect of introducing a *para*-methoxy (*p*-OMe) substituent on the phenyl ring at N4 (R substituent) on 3CLpro inhibitory activity (**THIA-6** vs. **THIA-4**; Table 2) was investigated by fitting **THIA-6** into the 3CLpro active site, based on docking results obtained for **THIA-4** (pharmacophore fitting). The calculation of the ligand-protein interactions for the resulting **THIA-6**/3CLpro complex revealed that the *p*-OMe group of the R substituent could establish favorable hydrophobic contacts with Pro^168^ in the S3 subsite (Appendix A). However, this substituent does not enhance compound selectivity, as it can also be accommodated within the active sites of Cathepsin L and Papain (see the following paragraph).

#### 2.5.3. Steered Molecular Dynamics Simulations

To elucidate the dynamic nature of ligand–protein interactions and validate docking results, Steered Molecular Dynamics (SMD) simulations were performed using the final docked complexes as starting conformations. SMD simulations of ligand unbinding from catalytic sites offer valuable insights into the molecular recognition process and can indirectly inform the binding mechanism [50]. In our case, however, we are not mimicking the unbinding process itself, but rather the reverse of the binding event, by guiding the ligand away from the catalytic site along a plausible dissociation pathway. This approach is particularly useful for exploring the conformational states and energetic routes that the ligand may traverse to reach the catalytic site [51].

The best pulling pathway was directed through the S_3_ subsite, the widest channel leading out of the catalytic pocket. For all compounds, the interaction energy with the protein decreased going from the starting (docked) to the final (unbound) conformation (Appendix A). At the end of the simulations, all compounds exceeded the 20 Å distance between S1 and the sulfur atom of Cys145 (Appendix A), which represents the threshold beyond which a ligand can be considered unbound from the protein site [52].

Importantly, the complexes retaining the BM-I binding modes resulted in the ones with the best interaction energy, further supporting the results of our docking studies, as reported in Figure 9.

SMD simulations showed that **THIA-2** is the most effective compound in maintaining the binding mode identified by docking studies (BM-I) with interaction energies ranging from –48.66 to –38.07 kcal/mol and the longest (65 ps) retention time of this binding mode among all THIA derivatives (Figure 9). This good fit with the catalytic site is in line with docking results and, as previously observed, could compensate for the low propensity to acquire electrons of **THIA-2** (the highest LUMO energy among THIAs).

Finally, local motions and conformational changes induced by THIA binding were assessed through the calculation of Root Mean Square Fluctuation (RMSF), by aligning each trajectory frame to the initial structure using all Cα atoms. Residues with RMSF values greater than 2 Å were considered to exhibit notable flexibility.

Results (Figure 10 and Appendix A) revealed that **THIA-4** displayed the highest fluctuation in the upper loop region (residues 44–54). This region plays a key role in stabilizing the active conformation of the catalytic residues as well as the ligand binding [53]

This result may account for the low inhibitory potency against 3CLpro of **THIA-4**, comparable to that of **THIA-2** and **THIA-8** (Table 2), despite the higher propensity to accept electrons (LUMO energy) compared to **THIA-2** and **THIA-8**.

#### 2.5.4. Investigation of the Molecular Bases of THIA Selectivity

To rationalize the selectivity profiles exhibited by the THIA derivatives against the tested cysteine proteases (Table 2), we conducted a comparative analysis between our computational findings and the experimentally resolved structures of the corresponding enzymes. Initially, we analyzed the size, shape, and relative orientation of the catalytic subsites of 3CLpro, PLpro, Papain, Cathepsin L, and Bromelain (Appendix A). Subsequently, the THIA scaffold was fitted into the active sites of PLpro (PDB ID: 6WX4), Papain (PDB ID: 6TCX), Cathepsin L (PDB ID: 3OF8), and Bromelain (PDB ID: 6YCG) by superimposing it onto the co-crystallized peptide ligands (pharmacophore fitting).

Papain, Cathepsin L, and Bromelain share a common fold topology and exhibit ~48% sequence similarity, allowing structural superimposition based on sequence alignment (Appendix A). In contrast, PLpro and 3CLpro show less than 30% sequence similarity with these proteases and cannot be structurally superimposed by their sequence alignment. However, PLpro retains a similar active site architecture and catalytic residue orientation (Appendix A), enabling its inclusion in the superimposition by aligning its catalytic triad (Cys^111^, His^272^, Asp^286^) with that of Cathepsin L (Cys^26^, His^164^, Asn^188^). 3CLpro, which differs significantly in both fold topology and catalytic residue orientation (Cys^145^, His^41^), was superimposed by aligning the backbone atoms of its co-crystallized peptide ligand (PDB ID: 7JKV) with those of the peptide ligand co-crystallized with PLpro (PDB ID: 6WX4).

This ligand-based structural alignment enabled a comparative analysis of the positioning of the subsites relative to the catalytic cysteine and the scissile peptide bond. While the S_1_ and S_2_ subsites were found to be roughly superimposed across all proteases, the S_3_ subsite exhibited distinct orientations. A schematic representation is reported in Figure 11.

Notably, the oxyanion holes of Papain, Cathepsin L, Bromelain, and PLpro are differently oriented compared to those of 3CLpro. In 3CLpro, it is located between the S_1_ and S_2_ subsites, whereas in the other proteases it is positioned within the S_1_ subsite, on the opposite side relative to the catalytic cysteine (Figure 11, Figure 12A–C and Appendix A). This implies a different spatial arrangement of the ligand-reacting atoms, as confirmed by the analysis of the X-ray structures. Accordingly, the fitting of the THIA skeleton into the active sites of Papain (PDB ID: 6TCX), Cathepsin L (PDB ID: 3OF8), Bromelain (PDB ID: 6YCG), and PLpro (PDB ID: 6WX4) places both the sulfur (S1) and nitrogen (N2) atoms in the S_1_ subsite, whereas in 3CLpro, N2 is located in the S_2_ subsite (Figure 12D–F and Appendix A).

In PLpro, the catalytic residues are located at the interface between the thumb (aa 62–178) and palm (aa 241–315) domains, and the S_1_ and S_2_ subsites are very narrow (Figure 8 A), in line with the substrate specificity of this enzyme at positions P_1_ and P_2_ where only glycine residues can be accepted [25,38,54]. Accordingly, the THIA derivatives cannot fit into the PLpro active site (Figure 12D). However, they could occupy the PLpro S_3_ and S_4_ subsites, thus closing the access to the catalytic site without forming a covalent bond with the catalytic Cys^111^, similar to what has been demonstrated for GRL0617 (IC_50_: 2.3 μM) and its derivatives, a class of potent SARS-CoV-2 PLpro allosteric inhibitors characterized by a structure similar to THIAs (Figure 13) [23,55,56].

In Figure 10A, the structural superimposition of **THIA-5** with a GRL0617 derivative in complex with PLpro (PDB ID: 7JIW) is reported; it is worth noting that **THIA-5** may interact with most of the residues also involved in the binding of the GRL0617 derivative (Appendix A). This accounts for the observed reversible Inhibition of PLpro by THIA compounds (Appendix A).

On the contrary, the active sites of Papain and Cathepsin L are less constrained (Figure 12C and Appendix A), allowing them to accommodate bulkier substrates, in line with the broader specificity of these proteases [8,57]. In particular, the catalytic cysteine is more solvent-exposed and accessible than in the other proteases, as shown by the solvent-accessible surface area (SASA) analysis (Appendix A). Accordingly, THIA derivatives—including those bearing a benzyl group such as **THIA-2**—can fit into the active sites of Papain and Cathepsin L (Figure 12F and Appendix A). As previously discussed, their different ability to inhibit these proteases appears to depend primarily on their distinct reactivity (E_LUMO_; Table 3).

The overall inactivity of THIA derivatives against Bromelain is likely due to the peculiar amino acid composition of its S_2_ subsite. Indeed, while the S_1_ subsite is strictly conserved, the S_2_ subsite is very different among Papain, Cathepsin L, and Bromelain (Appendix A). The S_2_ subsites of Papain and Cathepsin L are hydrophobic and, accordingly, bind to substrate/inhibitors presenting hydrophobic residues at P_2_. On the contrary, the S_2_ subsite of Bromelain is polar and negatively charged due to the presence of Asp^209^ and Glu^68^, in line with the ability of this enzyme to recognize substrates containing a positively charged residue at the P_2_ position [8,36]. The unique capacity of **THIA-7** to inhibit Bromelain (IC_50_ = 2.0 ± 0.1μM) could be due to the ability of the chlorine atom at the *para* position of the phenyl ring at N4 (R substituent) to establish a halogen bond with the Glu^68^ carboxylate group in the S_2_ subsite of this enzyme (Appendix A), as resulted from the calculation of ligand-protein interactions using a nonbonded geometric interaction criteria [47].

Our binding hypothesis could also account for the overall higher inhibitory activity of all THIA derivatives on Papain compared to Cathepsin L (Table 2). Indeed, the S_2_ subsite of Papain is characterized by less bulky residues compared to those present in Cathepsin L (Papain: Pro^68^, Val^157^ vs. Cathepsin L: Met^70^, Met^161^; Appendix A), leading to a better fit of the R substituent (Appendix A). In particular, the methoxyl group of **THIA-6** could establish a hydrogen bond with Ser^205^, a residue exclusive to Papain (subsite S_2_; Appendix A) in line with the highest Papain inhibitory activity of **THIA-6** among the tested compounds (Table 2).

#### 2.5.5. Covalent Docking Simulation

In order to obtain the molecular models of the covalent adducts of 3CLpro, Papain, and Cathepsin L with **THIA-3**, **-4**, **-6**, and **-8**, we investigated the ability of the catalytic cysteine-THIA covalent adduct (Figure 1) to adapt to the catalytic sites of the different proteases by applying the covalent docking procedure in AutoDock4 [58].

No specific starting ligand/protein complex was considered, and the adduct was randomly generated to obtain the best complex after product formation. Moreover, the conformation of the resulting adduct (resulting from the opening of the THIA ring) was allowed to change in order to find the best adaptation to the enzyme site.

Since **THIA-6**, **THIA-7**, and **THIA-10** exhibit stronger inhibitory effects (IC_50_ < 0.05) compared to the other THIAs with similar IC_50_ values, we focused on comparing the covalent binding of 3CLpro with **THIA-6** and **THIA-4**, as they differ only by the substitution of a p-methoxyphenyl group in **THIA-6** versus a phenyl group in **THIA-4** at the N4 substituent (R). Covalent docking revealed that, following reaction with the catalytic Cys^145^, both compounds occupy the S_1_–S_2_–S_4_ subsites and one hydrogen bond with the N^δ^ atom of His^41^. In this binding mode, the N4 substituent (R) is accommodated within the S1 subsite (Phe^140^, Leu^141^, Asn^142^, Met^165^, Glu^166^) while the N2 substituent (R′) bearing a p-methoxyphenyl group penetrates deeply into the hydrophobic S_2_ pocket (Met^49^, Asp^187^, Arg^188^), with the p-methoxy substituent extending additional interactions toward the S_4_ subsite (Met^165^, Glu^166^, Gln^189^ and Gln^192^) (Figure 14A).

Although both compounds adopt a similar positioning within the 3CLpro binding pocket, the p-methoxy substituent at the N4 position of **THIA-6** enabled additional interactions within the S_1_ subsite, including contacts with His^172^. This deep interaction with the S1 subsite allowed an improved fit across the S_2_–S_4_ subsites and conferred slight solvent accessibility compared to **THIA-4** (Figure 14B). It is noteworthy that the covalent docking differs from the docking complex shown in Figure 8 regarding the position of N4 substituents (R substituent) that occupy the S_3_ subsite in all complexes.

Next, we evaluated **THIA-4**, **THIA-6**, and **THIA-8** using covalent docking simulations with Papain adducts. These compounds were selected because, although they differ only at the N4 substituent, **THIA-6** exhibits a sevenfold higher inhibitory potency than **THIA-4** and a 240-fold increase compared to **THIA-8**. The covalent docking simulations further revealed that the N4 substituent (R) generally occupies the S_2_ subsite and the N2 substituent (R’) occupies the S_3_ subsite, except in the complex with **THIA-8**, where this orientation is inverted (Figure 15A).

The binding site of papain should not be divided simply into S_2_ (Val^133^, Val^157^, and Asp^158^) and S_3_ (Gly^66^ and Tyr^67^) subsites, but rather as a single large hydrophobic pocket encompassing both regions [59]. We observed that **THIA-4** and **THIA-6** displayed similar interactions of the p-methoxyphenyl group at the N2 substituent within the S_3_ subsite and showed greater occupancy of the S_2_ subsite core due to the phenyl substituent at N4 near the S_2_ residues. **THIA-6**, in particular, exhibited a higher number of interactions within the S_2_ subsite than **THIA-4** owing to the O-methyl group, including additional contacts with Phe^297^ and Trp^69^.

In contrast, the p-methoxyphenyl group at the N2 position of **THIA-8** was deeply inserted into the S_2_ subsite. In contrast, the cyclohexyl at the N4 substituent extended toward the S3 region, remaining distant from Gly66 and Tyr67, and instead being solvent-exposed (Figure 15B). Based on the structural features of both the receptor and the ligands, we hypothesize that the papain binding pocket is not deep enough to accommodate the bulky cyclohexyl group of **THIA-8**. This steric limitation likely results in an incorrect geometric alignment of the warhead within the active site, redirecting the binding orientation and forcing the N2 substituent into the S_2_ subsite while leaving the warhead poorly positioned in the S_3_ pocket.

Cathepsin L (CatL) contains a deep and narrow hydrophobic S2 subsite defined by the side chains of Leu^69^, Met^70^, Ala^135^, and Met^161^, together with the backbone atoms of Met^161^, Asp^162^, His^163^, and Gly^164^, highlighting their essential roles in substrate specificity [60], as exemplified in the Cat L–E64d complex (PDB ID: 7ZXA). Moreover, the restricted architecture of the S_2_ pocket is reinforced by Asp^160^, Met^161^, Asp^162^, and Ala^214^, which collectively form a continuous steric hindrance barrier that prevents further extension of the S_2_ subsite [61]. In this context, the bulky cyclohexyl moiety may fit the S_2_ subsite, similar to that of Triazine, which addresses the hydrophobic S_2_ pocket [62] (PDB ID: 4AXM) (Figure 16A).

The distinct hydrophobicity, especially in the S_2_ and S_3_ subsites (Gly^61^, Glu^63^, Gly^67^, Gly^68^), favors different related hydrophobic side chains of the compounds, and hydrogen bond hotspots across the subsites of Cat L include Asp^162^ and Gly^68^ [63]. The calculated adducts of Cathepsin L with **THIA-3** and **-8** show that in the complex **THIA-3**—Cathepsin L, the phenyl substituent at N4 is located in the S_1_ subsite, and the phenyl group of N2 points to the entrance of the S_2_ subsite, having an H-bond with Gly^68^ and Asp^162^. In the complex **THIA-8**—Cathepsin L, the N4 substituent (cyclohexyl; R) points to S_2_ containing an H-bond with Gly^68^ and Gly^164^, and the N2 substituent is solvent-exposed (Figure 16B).

Although the cyclohexyl moiety fits well into the S_2_ pocket of Cat L, the p-methoxyphenyl group at the N2 substituent of **THIA-8** exhibits suboptimal accommodation in the S_3_ subsite. Possibly the relatively bulky volume of this moiety, combined with the narrow geometry of the Cat L S_3_ pocket, which leads to a steric clash with the pocket wall, as indicated by Glu^63^, results in partial solvent exposure. In contrast, the smaller substituent composition of **THIA-3** favors more extensive interactions within the S_1_ subsite and achieves a better fit inside the S_3_ subsite (Figure 16C).

It is noteworthy that the presence of the bulky side chain of Met^70^ and Met^161^ in the S_2_ subsite (Appendix A) restricts its occupancy that may account for the overall weaker inhibitory activity of THIA derivatives on Papain compared to the other two proteases (3CLPro and Cathepsin L; Table 2), particularly **THIA-1**, **THIA-2**, and **THIA-8**, characterized by bulkier substituents at N4, compared to the other THIAs.

## 3. Materials and Methods

### 3.1. Chemistry

Anhydrous dimethyl sulfoxide (DMSO) and acetonitrile (ACN) were from Sigma-Aldrich (St. Louis, MO, USA). All buffer salts were reagent-grade and purchased from Fisher Scientific (Pittsburgh, PA, USA) or Sigma-Aldrich. The resins and all peptide synthesis-grade reagents (N-methylpyrrolidone (NMP), N-methylmorpholine (NMM), dichloromethane, piperidine, trifluoroacetic acid (TFA), anisole, thioanisole, and triisopropylsilane) and 5-carboxyfluorescein were purchased from Sigma (Saint-Quentin Fallavier, France).

The FRET peptides were obtained by solid-phase synthesis, using the Fmoc (N-(9-fluorenyl)-methoxycarbonyl) from Novabiochem (Darmstadt, Germany), on an automated multiple peptide synthesizer (PSSM-8 system; Shimadzu, Tokyo, Japan), as previously described [6,28]. Syntheses were carried out on a NovaSyn^®^ TGR resin (Millipore, Burlington, MA, USA), using HBTU (N, N’-tetramethyl-O-benzotriazo-1-yluronium tetrafluoroborate)/HOBt (1-hydroxybenzotriazole) as coupling reagent. The removal of peptide from the resin was accomplished with TFA: thioanisole: 1,2-ethanedithiol: water (85:5:3:7). Semipreparative HPLC purified peptides on a C18 column (Econosil™; Fisher Scientific, Pittsburgh, PA, USA) and their molecular weight and purity (>95%) checked by reverse-phase chromatography and by mass spectrometry (Shimadzu, Tokyo, Japan). Stock solutions of peptides were prepared in ACN, and the concentration was measured spectrophotometrically using the Dnp molar extinction coefficient of 17,300 M^−1^ cm^−1^ at 365 nm.

All the 1,2,4-thiadiazolidin-3,5-diones were synthesized through oxidative condensation of the appropriate isothiocyanates and isocyanates in the presence of SO_2_Cl_2_, and then purified and characterized as previously reported [3]. **THIA-3**, selected as a reference compound because of its simplicity, was produced on a larger scale. As an alternative to crystallization, employed in the previously described procedure, and in order to achieve a higher degree of purity associated with high recovery efficiency of the desired product, **THIA-3** was purified at >99% purity by direct phase flash chromatography (5–40% ethyl acetate/n-hexane) on a Biotage Selekt instrument (Biotage, Uppsala, Sweden) equipped with a Sfär Duo 10 g column.

### 3.2. Enzymes

Recombinant human Cathepsin L was expressed in *Pichia pastoris* following the previously described procedures [64]. The enzyme and substrate were used under the following conditions: 100 mM sodium acetate buffer containing 1 mM EDTA, pH 5.0, with substrate Z Phe-Arg-MCA.

Papain and Bromelain were purchased from Sigma. Papain was previously activated at 25 °C in a degassed 100 mM sodium phosphate, 1 mM EDTA, pH 6.0 buffer containing 5 mM DTT (dithiothreitol) for 10 min, as previously described [65]. To remove the DTT excess, a 1 mL solution of activated Papain was loaded on a 3 mL syringe with 1 mL Sephadex G25 size exclusion resin (Amersham Pharmacia, Amersham, UK). The enzyme was eluted with the same assay buffer, and its activity was monitored using Z-Phe-Arg-MCA as the substrate.

Recombinant SARS-CoV-2 proteases 3CLpro were expressed from E. coli as previously described [17,66] with some modifications. Briefly, the culture was pre-cultured in LB medium with ampicillin (100 μg/mL) and then transferred to ZYM 5052 self-inducing medium, shaking at 37 °C until OD 600 0.7–0.8, when the protein expression was conducted at 18 °C for 20 h at 180 rpm. Afterward, the cells were centrifuged, the bacterial pellet was resuspended, and the cells were lysed using French Pressure (3 times at 7000–10,000 psi) and 20 μL of Benzonase Nuclease (Millipore). The supernatant containing the soluble protein was then loaded onto a nickel affinity column. To cleave the His-6 Tag from the purified protein, the Pierce kit™ HRV3C Protease Solution, also known as PreScission Protease (Thermo Scientific, Waltham, MA, USA), was used according to the manufacturer’s instructions. The reaction was transferred into a dialysis cassette (Slide-A-Lyzer dialysis cassette, 10 K, Thermo Scientific) and dialyzed against buffer C (20 mM Tris-HCl, pH 7.8, containing 150 mM NaCl and 1 mM DTT) at 4 °C for 20 h. In order to remove the HRV 3C protease, His6-tag, and eventual non-cleft 3CLpro His-6-tag, the dialysis solution was applied to a GST Trap FF (GE Healthcare, Chicago, IL, USA) column connected to a nickel column HisTrap FF (GE Healthcare) [17]. The purified 3CLpro was stored in buffer D at 4 °C. Recombinant SARS-CoV-2 proteases PLpro were also expressed from E. coli, as previously described [67].

The activities of Cathepsin L and Papain were assessed using Z-Phe-Arg-MCA. Cathepsin L was assayed in the standard reaction buffer, 100 mM sodium acetate, 1 mM EDTA, pH 5.0, degassed, and Papain was assayed in 100 mM sodium phosphate, 1 mM EDTA, pH 6.0, degassed.

Bromelain substrate was Z Arg-Arg-MCA in 50 mM sodium acetate, 1 mM EDTA, pH 5.0. The SARS-CoV-2 3CLpro and PLpro assays were conducted using the FRET-based substrates, Abz SAVLQSGFRK(Dnp)NH2 and Abz TLKGGAPIKQ-EDDnp, respectively, as reference [16,68].

#### 3.2.1. Determination of IC_50_

The assays of proteases were performed as described above, and the enzyme inhibition was expressed as the THIAs concentrations, causing a 50% decrease in enzyme activity (IC_50_). We calculated them by nonlinear regression, dose response curves using THIAs at different concentrations, and the data were analyzed in Grafit 5.0 software (Erithacus, London, UK) using Equation (1): [y = 100%/1 + (x/IC_50_)s], where 100% is the fitted uninhibited value and s is a slope factor. The equation assumes that y falls with increasing x. All the assays were done using a SpectraMaxM2e (Molecular Devices, San Jose, CA, USA) with 320 nm excitation and 420 nm emission for FRET peptides and 360 nm excitation and 480 nm emission for MCA substrates.

#### 3.2.2. Statistical Analysis

IC_50_ data are reported as mean ± S.D. Statistical differences between groups were evaluated using Student’s *t*-test. *p* < 0.05 was considered significant.

### 3.3. Molecular Modeling Studies

Molecular modeling calculations were performed on a CPU/GPU hybrid High-Performance Computing Cluster (10 Twin servers (DoIT Systems Srl, Torino, Italy), for a total of 560 Intel^®^ Xeon^®^ Gold processors (128 GB RAM), 64 AMD^®^ EPYC^®^ processors, and 2 GPU NVIDIA^®^ Tesla^®^ V100). The molecular modeling graphics were carried out on a personal computer equipped with an Intel(R) Core (TM) i7-8700 processor.

#### 3.3.1. Conformational Analysis

The THIA derivatives were built using the Small Molecule tool of Discovery Studio 2023 (Dassault Systèmes BIOVIA, San Diego, CA, USA). Atomic potentials and charges were assigned using the cff force field [69]. The conformational space of the compounds was sampled using the random search algorithm, Boltzmann Jump, for the random generation of a maximum of 300 conformations. Applying this method, each random perturbation is either accepted or rejected according to the Metropolis selection criterion with a ratio according to the Boltzmann distribution (T = 300 K). Finally, an energy threshold value of 10^6^ kcal/mol was used as the selection criterion. The generated structures were then subjected to Molecular Mechanics (MM) energy minimization (ε = 8 × r) until the average RMS gradient during a cycle of minimization was less than 0.001 kcal/Å, using Conjugate Gradient [70] as minimization algorithm. The resulting conformers were ranked by their potential energy values (i.e., energy difference from the global minimum (ΔE_GM_)) and classified according to the values of the torsion angles.

All the MM energy conformers were then subjected to DFT calculations (Gaussian 16 package) [71]. All structures were fully optimized at the B3LYP/6–311++G(d,p) using the conductor-like polarizable continuum model (C-PCM) [41]. The C-PCM method allows the calculation of the energy in the presence of a solvent. In this case, all structures were optimized as a solute in an aqueous solution. In order to characterize every structure as a minimum, a vibrational analysis was carried out (keyword = freq). The RMS force criterion was set to 3 × 10^−4^ a.u. Molecular orbitals have been calculated using the natural bond orbital (NBO) method [72]. The resulting DFT conformers were ranked by their potential energy values (i.e., ΔE from the global energy minimum) and classified according to the values of the torsion angles.

#### 3.3.2. Docking Studies on 3CLpro

The experimentally determined structure of 3CLpro with the best resolution (1.25 Å) in complex with the covalent peptidomimetic inhibitor GRL-2420 (PDB ID: 7JKV) [73] was selected as the starting structure for the docking studies.

The coordinates of 7JKV were downloaded from the Protein Data Bank (PDB; https://www.rcsb.org/, accessed on 1 April 2025). The structure was refined by using the “Prepare protein” tool of Discovery Studio 2023 (Dassault Systèmes BIOVIA, San Diego, CA, USA). In particular, the alternate conformations of side chains of the residues A: Asp^245^, A: Thr^19^6, A: Val^186^, A: Met^165^, A: Met^162^, A: Ser^139^, A: Val^125^, A: Ser^121^, B: Ser^254^, B: Thr^226^, B: Thr^196^, B: Met^162^, B: Val^125^, B: Leu^87^, B: Val^86^, B: Arg^76^, B: Ser^62^ were deleted. The hydrogens were added to the structure, assuming a pH of 7.2 by using the “Calculate Protein Ionization and Residue pK” command, while the N-terminal and C-terminal residues were considered charged. The inhibitor of 7JKV was removed, and atomic potentials and partial charges were assigned using the cff force field.

Docking studies were conducted on **THIA-2**, **THIA-4**, **THIA-7**, **THIA-8**, and **THIA-10** using the LigandFit docking procedure, a shape-based grid docking combined with MM energy minimization (cff forcefield; Dassault Systèmes BIOVIA, San Diego, 2023) [43]. To increase the variance of the starting complexes (i.e., starting ligand poses), the resulting DFT conformers of **THIA-2** (four conformers), **THIA-4** (two conformers), **THIA-7** (two conformers), **THIA-8** (four conformers) and **THIA-10** (one conformer) were placed in the catalytic site of 3CLpro. According to the molecular mechanism of proteolysis of 3CLpro and considering the positioning of substrate [42], each starting conformer was placed in the catalytic site of 3CLpro, positioning the R substituent into the S_1_ subsite, the R’ substituent into the S_2_ subsite and the N2 in the oxyanion hole [74].

Starting complexes were then subjected to docking studies. Docking parameters (i.e., RMS threshold for ligand-site matching, dimensions of the grid, dielectric constant) were appropriately tailored to the system on the basis of a series of “test docking” calculations performed on 7JKV. In particular, the position of the peptidomimetic inhibitor was manually modified by placing its substituents in different subsites compared to the starting X-ray structure. Then, the parameters able to reproduce the X-ray binding mode of GRL-2420 (PDB ID: 7JKV) were chosen.

During the docking calculations, the protein was held rigid, while the ligands were allowed to move by a random combination of translation and rotation changes to sample their orientation within the defined binding domain area without changing their conformation. According to the results of the structural and bioinformatics analysis, to properly investigate the binding mode of THIA, for each starting conformation of the ligands, the binding domain area was defined using a binding site sphere of 12 Å around the mass center of the inhibitor GRL-2420, covering the S_1_-S_5_ and S_1’_-S_2’_ subsites (Volume: 522.5 Å^3^). Moreover, in order to investigate the first approach of our compounds to the catalytic site before the nucleophilic attack, a tethering restraint was applied on the hydrogen bond between the catalytic residues Cys^145^ and His^41^ (constrained within 3.0 Å using a force constant of 100 (kcal/mol)/Å). Firstly, random orientations of the starting conformation in the binding domain area were generated using an RMS threshold for ligand-site matching of 10 Å. To increase the sampling of ligands in the binding site, a softened Lennard-Jones 9–6 potential was used for calculating the van der Waals energy in the energy calculation.

Moreover, in order to ensure that the energy for ligand atoms outside of the defined binding domain area was calculated, the dimensions of the grid were extended to 3.0 Å from the defined binding domain area. Then, in order to refine the resulting poses, these latter were minimized using steepest descent (1000 steps) and Broyden–Fletcher–Goldfarb–Shanno (BFGS; 2000 steps) as minimization algorithms (ε = 80 × r). The poses were selected using an RMS threshold of 1.50 Å. For each ligand conformation, 20 poses were generated.

The generated complexes were then subjected to MM energy minimization until the average RMS gradient during a minimization cycle was less than 0.1 kcal/Å, using the Conjugate Gradient algorithm (cff; ε = 80 × r). During the minimization, all the residues of 3CLpro were left free to move. In order to investigate the first approach of our compounds to the catalytic site before the nucleophilic attack, a tethering restraint was applied on the hydrogen bond between the catalytic residues Cys^145^ and His^41^ (constrained within 3.0 Å using a force constant of 100 (kcal/mol)/Å). In order to preserve the THIA conformation obtained from DFT calculations, dihedral restraints were applied on the torsion angles using a force constant of 100 kcal/mol.

For each docking calculation, the complexes, characterized by the distance between the S1 sulfur atom of THIA and the sulfur atom of Cys^145^ < 4 Å, were selected. The structures were ranked by their ligand-protein non-bond interaction energy values (vdW and electrostatic energy contribution; atom-based method; Nonbond List Radius = 15.0; Higher Cutoff Distance = 14.0; Lower Cutoff Distance = 11.0; ε = 2 × r; Simulation tool, Discovery Studio 2023) and classified by the positioning of the R and R’ substituents in the 3CLpro subsites, defining the different ligand binding modes: BM-I, BM-II, BM-III, and BM-IV. The orientation and the distance (<4.5 Å) of the N2 nitrogen with respect to the hydrogen catalytic cysteine were checked, and the binding mode BM-I was accordingly selected.

#### 3.3.3. Analysis of the Selected Docked Complexes

The protein structural quality of the selected complexes was assessed using Procheck structure evaluator software, version 3.5.4 [75] and compared to that of the reference structure (PDB ID: 7JKV). Structural analyses of the selected docked complexes were performed using Macromolecules and Receptor-Ligand Interaction tools of Discovery Studio 2023 (Dassault Systèmes BIOVIA, San Diego, 2023) and taking into account the results of the structural and bioinformatics analysis. Finally, the ligand-protein non-bond interaction energy (i.e., the van der Waals term and the electrostatic term; kcal/mol) was calculated, and the contribution of each protein residue within 5 Å from any ligand atom was assessed (vdW and electrostatic energy contribution; atom-based method; Nonbond List Radius = 15.0; Higher Cutoff Distance = 14.0; Lower Cutoff Distance = 11.0; ε = 2 × r; Simulation tool, Discovery Studio 2023). The residues with a favorable non-bond interaction ≥ 1 kcal/mol were considered key interaction residues. The types of ligand-protein interactions were evaluated using the nonbonded geometric interaction criteria (Receptor-Ligand Interaction tool; Discovery Studio 2023) [49].

#### 3.3.4. Steered Molecular Dynamics

To validate docking results, the selected docked complexes were subjected to Steered Molecular Dynamics (SMD). As in the docking studies, SMD was employed to explore the initial approach of the compounds to the catalytic site prior to nucleophilic attack. Accordingly, during all the following calculations, a tethering restraint was applied on the hydrogen bond between the catalytic residues Cys^145^ and His^41^ (constrained within 3.0 Å using a force constant of 100 (kcal/mol)/Å). To preserve the THIA conformation obtained from DFT calculations, dihedral restraints were applied on the torsion angles using a force constant of 100 (kcal/mol)/Å.

The starting structures were solvated into a cubic periodic box using the TIP3P water model [76]. The box edges were set at least 12 Å around the protein, and periodic boundary conditions (PBC) were applied. To keep the electroneutrality of the system, eight Cl^−^ ions were added. All structures were parametrized using the CHARMM force field [77]. To eliminate steric clashes, the solvated systems were subjected to energy minimization using the Conjugate Gradient algorithm [70] until the average RMS gradient was less than 0.1 kcal/mol·Å. Subsequently, solvent molecules were equilibrated for 50 ps under an isothermal–isobaric (NPT) ensemble, with protein and ligand atoms held fixed. Temperature was maintained at 298 K using the Nosé–Hoover thermostat [78] and pressure was controlled at 1 atm using the Langevin piston method [79]. During the simulation, the SHAKE algorithm [80] was utilized to constrain the hydrogen-involved bonds, and the particle mesh Ewald method [81] was used for the calculation of electrostatic contributions to the nonbonded interactions, with the nonbonded cutoff distance set to 12 Å. The final equilibrated structures were used as starting points for SMD simulations. Each compound underwent a 500 ps SMD run with a 1 fs time step. Based on the active site geometry and the positioning of THIA derivatives after docking studies, the pulling pathway was directed (1) through the S_3_ subsite, the widest channel leading out of the catalytic pocket or (2) through the upper and lower loops as defined in Figure 10. Steering forces were applied between the sulfur atom of Cys^145^ and either the para-position carbon of the R substituent (**THIA-4**, **THIA-7**, **THIA-8**, **THIA-10**) or the methylene carbon (**THIA-2**). A spring constant of 350 pN/Å (equivalent to 5 kcal/mol·Å) [50,51] and a pulling speed of 0.04 Å/ps were applied. Trajectories were saved every 5 ps for subsequent analysis.

Structural analyses of the resulting conformations were performed using Receptor-Ligand Interaction and Simulation tools of Discovery Studio 2023 (Dassault Systèmes BIOVIA, San Diego, 2023). The ligand-protein non-bond interaction energy (i.e., the van der Waals term and the electrostatic term; kcal/mol) was calculated (vdW and electro-static energy contribution; atom-based method; Nonbond List Radius = 15.0; Higher Cut-off Distance = 14.0; Lower Cutoff Distance = 11.0; ε = 2 × r; Simulation tool, Discovery Studio 2023) including the water molecules involved in the interaction between THIA and the protein binding site.

The types of ligand-protein interactions were evaluated using the nonbonded geometric interaction criteria (Receptor-Ligand Interaction tool; Discovery Studio 2023) [49]. Since the CFF force field used in docking is not compatible with PBC [82], nonbonded interaction energies for the docked complexes were recalculated using the CHARMM force field to ensure consistency with the SMD simulations. THIA compounds were considered to preserve the BM-I binding mode when the following criteria were met: (i) the distance between the S1 sulfur atom of THIA and the sulfur atom of Cys^145^ was less than 4 Å; (ii) the N2 nitrogen was within 4.5 Å of the catalytic cysteine hydrogen; (iii) N2 and N4 substituents occupied the S_2_ and S_3_ subsites; (iv) hydrogen bond with Gln^189^ was maintained. Finally, Root Mean Square Fluctuation (RMSF) values were calculated for each residue. RMSF was computed by aligning each trajectory frame to the initial structure using all Cα atoms. Residues with RMSF values greater than 2 Å were considered to exhibit notable flexibility.

#### 3.3.5. Bioinformatics and Structural Analysis

The experimentally determined structures of (i) 3CLpro (PDB IDs: 7JKV and 7N89), (ii) PLpro (PDB IDs: 6WX4 and 7JIW), (iii) Papain (PDB IDs: 6TCX), (iv) Cathepsin L (PDB ID: 3OF8) and, (v) Bromelain (PDB ID: 6YCG) were downloaded from the Protein Data Bank (PDB; http://www.rcsb.org/pdb/).

All structures were parametrized using the CHARMM force field [77] and analyzed by using Macromolecules, Simulation, and Receptor-Ligand Interaction tools of Discovery Studio 2023 (Dassault Systèmes BIOVIA, San Diego, 2023).

To perform the analysis, the structures were superimposed using the following procedure. A secondary structure sequence alignment among Cathepsin L (PDB ID: 3OF8), Papain (PDB ID: 6TCX), and Bromelain (PDB ID: 6YCG) was performed using the Align123 algorithm [83] (Secondary structure: DSC Pairwise Alignment method: Slow; Scoring Matrix: BLOSUM; Gap Open Penalty: 10; Gap Extension Penalty: 0.05; Macromolecules tool; Discovery Studio 2023). The structures were superimposed by fitting all Cα atoms of the aligned residues, and the RMSD value of each Cα pair was calculated; then, a final superimposition was performed using only the Cα pairs with an RMSD < 1Å. The PLpro structure (PDB ID: 6WX4) was added to the superimposition by fitting the Cα atoms of its catalytic residues (Cys^111^, His^272^, and Asp^286^) on those of Cathepsin L (Cys^26^, His^164^, and Asn^188^). Finally, the 3CLpro structure (PDB ID: 7JKV) was added to the superimposition by fitting the backbone atoms of the P_1_ and P_2_ residues of the co-crystallized peptide ligand (GRL-2420) with those of the peptide ligand (VIR251) of PLpro (PDB ID: 6WX4). The other 3CLpro (PDB ID: 7N89) and PLpro (PDB ID: 7JIW) structures were also included and superimposed on 7JKV and 6WX4, respectively, according to their secondary structure sequence alignment using the same procedure reported above for Cathepsin L, Papain, and Bromelain.

The DFT conformers of THIA derivatives were fitted into the active site of PLpro, Cathepsin L, Papain, and Bromelain in order to properly orient S1 and N2 with respect to the catalytic dyad (Cys and His; oxyanion hole). The global minimum DFT conformer of **THIA-2** and **THIA-3** was superimposed on the peptide inhibitor VIR251 in complex with SARS-CoV-2 PLpro (PDB ID: 6WX4) by fitting: (i) the centroid of phenyl ring at N2 on the centroid of methyl ester group (subsite S_1_); (ii) the centroid of phenyl ring at N4 on the Cα of glycine (subsite S_2_), and (iii) the sulfur atom on the carbon atom covalently linked to the catalytic cysteine. The lowest energy DFT conformers (i.e., conformers 1A/B) of **THIA-2**, **THIA-6**, and **THIA-7** were superimposed on Phe-Tyr(OBut)-COCHO inhibitor in complex with Cathepsin L (PDB ID: 3OF8) by fitting: (i) the centroid of phenyl ring at N2 on the centroid of the side chain of tyrosine (subsite S_1_); (ii) the centroid of phenyl ring at N4 on the centroid of the side chain of phenylalanine (subsite S_2_), and (iii) the sulfur atom on the carbon atom covalently linked to the catalytic cysteine. According to this latter superimposition, **THIA-6** and **THIA-7** were also fitted into the active site of Papain and Bromelain.

The global minimum DFT conformer of **THIA-5** was fitted into the allosteric site of PLpro by superimposition with the noncovalent inhibitor (PDB ID: 7JIW) (Molecular Overlay; 50% steric, 50% electrostatic components; Discovery Studio 2023).

In all the obtained complexes, the number and types of ligand-protein interactions were evaluated using the nonbonded geometric interaction criteria (Receptor-Ligand Interaction tool; Discovery Studio 2023).

The surfaces of subsites S_1_, S_2,_ and S_3_ were created using the residues within 5 Å from any atom of the P_1_, P_2,_ or P_3_ substrate/inhibitor residues (Receptor-Ligand Interaction tool; Discovery Studio 2023). Solvent accessible surface area (SASA) calculations were performed using Discovery Studio 2023. The SASA of the catalytic cysteine sulfur atom was calculated for the following experimentally determined structures of 3CLpro (PDB IDs: 7JKV), PLpro (PDB IDs: 6WX4), Papain (PDB IDs: 6TCX), Cathepsin L (PDB ID: 3OF8), and Bromelain (PDB ID: 6YCG).

#### 3.3.6. Covalent Docking

Previously reported structures of 3CLpro (PDB: 7SF1), Papain (PDB: 1PE6), and Cathepsin L (PDB: 7W34) were adapted for covalent docking using PyMOL (version 3.0.3). Covalent inhibitors were present in these structures. Ligands, ions, and water molecules were removed, and hydrogens were added to the receptors using PyMOL. **THIA-3**, **-4**, **-6**, and **-8** were designed, and their topology parameters were generated via the SwissParam [84] and LigParGen [85] web servers. The inhibitors were then converted into mol2 format using Openbabel (version 3.1.1) for covalent docking. Covalent docking of the hypothetical Cysteine protease-THIA complexes was performed with the flexible side-chain method in AutoDock4 to predict the binding modes of the covalent complexes. Docking simulations utilized the default settings of the Lamarckian Genetic Algorithm, and ligand docking conformations were ranked based on a semi-empirical force field-derived scoring function [86]. Manual iterative adjustments of the covalent binding were subsequently performed in Coot [87] using the docked protease-inhibitor models. Visualization and preparation of structural figures were carried out using Schrödinger Maestro v. 14.5] and PyMOL (https://pymol.org/2/, accessed on 1 September 2025). Below, we present the structures of the simulation for covalent adduct formation between 3CLpro, Papain, and Cathepsin L with **THIA-3**, **-4**, **-6**, and **-8**.

## 4. Conclusions

The diversity among the ten THIA compounds assayed is still limited, which hinders a comprehensive analysis of the factors involved in the reactions studied, as well as other structures we plan to synthesize based on the data reported in this paper. However, we demonstrate that 1,2,4-thiadiazolidin-3,5-diones (**THIA-1**–**10**) can act as selective covalent inhibitors of specific cysteine proteases, including SARS-CoV-2 3CLpro, with varying degrees of activity also observed against reference enzymes such as Papain and Cathepsin L. Computational studies provided support for the experimental findings and offered a rationale for their selectivity profiles. The inhibition mechanism followed a two-step irreversible bimolecular process, but was reversible in the presence of reducing agents. The inhibitory activity seems to depend not only on the electrophilic reactivity of the THIA core, which enables covalent binding to the catalytic cysteine, but also on the conformational features of the scaffold and the ability of the R and R′ substituents to direct the compound toward specific protease subsites, promoting productive hydrophobic and electrostatic interactions. The most potent inhibitors (**THIA-6**, **-7**, and **-10**) exhibited nanomolar IC_50_ values against 3CLpro, while **THIA-1**, **-2**, and **-8** displayed enhanced selectivity. The benzyl or cyclohexyl group at the N4 nitrogen favored selectivity toward 3CLpro, whereas para-halogenated phenyl substituents increased overall potency. In the case of Cathepsin L, the reduced activity may be attributed to steric hindrance within the S_2_ subsite. The observed reversible inhibition of PLpro is likely due to the inaccessibility of subsites S_1_ and S_2_; the good structural overlap between the non-covalent inhibitor GRL0617 and **THIA-5** in complex with PLpro supports the putative binding of both compounds to the same allosteric site.

The dual functionality of THIA derivatives—as both H_2_S-releasing agents and covalent inhibitors—justifies the development of multifunctional therapeutic agents targeting cysteine proteases involved in viral replication and inflammatory processes.

## 5. Future Directions

We will focus on the design and synthesis of new THIA derivatives and on their encapsulation (to protect the reaction from reducing agents, such as glutathione) for biological assays. Based on our experience in the field [88], we will investigate THIA compounds also as inhibitors of human cysteine proteases (cathepsins B, K, L, and S) and cysteine proteases associated with parasitic infectious diseases, including cruzipain from *Trypanosoma cruzi* (Chagas disease), falcipain-2 from *Plasmodium falciparum* (malaria), and CPB2.8DCTE cysteine proteinase from *Leishmania mexicana* (leishmaniosis).

## Data Availability

The original contributions presented in this study are included in the article and Appendix A. Further inquiries can be directed to the corresponding authors.

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
