# Peer review of "1,2,4-Thiadiazolidin-3,5-Diones as Inhibitors of Cysteine Proteases"

_molecules, 2025, doi:10.3390/molecules30193896_

Round 1

Reviewer 1 Report

Comments and Suggestions for Authors

In this manuscript, the authors selected a series of 1,2,4-thiadiazolidin-3,5-dione (THIA) analogs and evaluated them in four cysteine protease enzyme assays. Based on assay results such as IC₅₀ values, and supported by molecular modeling studies—including both regular and covalent docking as well as ligand conformation analyses—the authors investigated the molecular basis of THIA analog activity to inform further scaffold modification. The study is overall well designed and well written. However, some minor revisions should be addressed before acceptance for publication:

  1. It is recommended to add an additional figure illustrating the S1, S2, and other subsites of each cysteine protease, better if in a comparative manner. This would help readers unfamiliar with the structural biology of cysteine proteases better understand the structural characteristics of these enzymes. A 2D schematic diagram would be nice and could provide readers with a more intuitive representation.
  2. The authors selected 10 THIA analogs for the study. However, the structural diversity among these analogs seems limited. This lack of diversity may narrow the scope of conclusions drawn. While expanding the number of analogs in the current work may not be possible, increasing chemical diversity in the future studies is recommended to better guide modification strategies.
  3. The authors mentioned that some THIA analogs are unstable in DMSO containing 1% water yet appear stable in acetonitrile with water or even in water-based solvents. It would be interesting to explore the mechanism behind this instability in DMSO–water mixtures. Additionally, it is also curious to know whether dry DMSO was tested for stability assessment.
  4. Typographical errors: In lines 117 and 120, “SARS-CoV” should be corrected to “SARS-CoV-2.”

Author Response

Comment 1. It is recommended to add an additional figure illustrating the S1, S2, and other subsites of each cysteine protease, better if in a comparative manner. This would help readers unfamiliar with the structural biology of cysteine proteases better understand the structural characteristics of these enzymes. A 2D schematic diagram would be nice and could provide readers with a more intuitive representation.

Response -According to the reviewer’s suggestion, a new 2D schematic figure illustrating the catalytic residues (i.e., Cys and His), the S1, S2, and S3 subsites, as well as the substrate, has been added on page 20 (i.e., Figure 9).

Comment 2. The authors selected 10 THIA analogs for the study. However, the structural diversity among these analogs seems limited. This lack of diversity may narrow the scope of conclusions drawn. While expanding the number of analogs in the current work may not be possible, increasing chemical diversity in the future studies is recommended to better guide modification strategies.

Response -According to the suggestion of the reviewer a new paragraph entitled “Future directions” has been added at page 33.

      Comment 3 - The authors mentioned that some THIA analogs are unstable in DMSO containing 1% water yet appear stable in acetonitrile with water or even in water-based solvents. It would be interesting to explore the mechanism behind this instability in DMSO–water mixtures. Additionally, it is also curious to know whether dry DMSO was tested for stability assessment.

Response: The study of the mechanism of THIA compounds decomposition is, in fact, fascinating; however, as reported in the main text, it is out of the focus of this paper. However, it is relevant to call attention to this observation, since DMSO is already considered a reagent ( see reference -28: Tashrifi, Z and cols. Advanced Synthesis & Catalysis 2019, 361 doi.org/10.1002/adsc.20190102). We observed stability of THIA-3 in dry DMSO, but for a period no longer than 3 h. In the experimental conditions, the tested solution may have absorbed atmospheric water. DMSO is very hygroscopic.  

Comment 4. Typographical errors: In lines 117 and 120, “SARS-CoV” should be corrected to “SARS-CoV-2.”

Response These typographical errors have been corrected.

Reviewer 2 Report

Comments and Suggestions for Authors

The manuscript is suitable for acceptance after addressing the following comments.

  1. Compare the known cysteine protease inhibitors with THIA derivatives as controls in the inhibition studies. Positive and negative controls are missing or not described
  2. The papain Ki​ and IC₅₀ data presented in the manuscript are difficult to interpret. The relationship between binding affinity (Ki) and inhibition concentration (IC₅₀) is not clearly explained or correlated. Please clarify this distinction and provide context to help readers understand the data.
  3. Inhibitor studies are performed at Different pH, papain in sodium phosphate (pH 6.0) Cathepsin in Sodium acetate (pH 5.0), 3Clpro in Tris (7.5), why given the unstability of THIA derivatives?
  4. Provide a good resolution image for Figure S4 and mention the number of replicates.
  5. The degradation studies of THIA derivatives are interesting and well described; however, the reason for the rapid degradation of these compounds is not explained. Is the degradation primarily due to pH effects, hydrolysis, or another mechanism? Please clarify why THIA-4 and THIA-9 exhibit complete degradation within 24 hours in DMSO. Additionally, THIA-1 and THIA-8 display three peaks at 12 hours with retention times between 7.5 and 13 minutes—please explain the origin of these peaks in the manuscript. For clarity, include a table summarizing the percentage degradation of each derivative over time.
  6.  The manuscript comments on the effect of R and R₂ substituents on the potency and selectivity of THIA derivatives; however, only a limited number of compounds were investigated, which restricts the strength of the conclusions that can be drawn. Given the computation resources and availability, it would be valuable to design and perform molecular docking studies on a broader range of THIA derivatives.
  7. Mention the binding velocity for Z-phe Arg-MCA, Abz-SAVLQSGFRK(Dnp)NH2 substrates with respective enzymes? how stable are these peptides in assay conditions? And selectivity towards other proteases.
  8. In the methods section, preactivation of the cysteine protease (CP) with DTT is described. Please specify the concentration of DTT used and the incubation time for this step. Additionally, considering that DTT requires extra steps such as removal by gel filtration to prevent interference in downstream assays, why was TCEP not used as an alternative reducing agent?
  9. Compare and discuss the docking scores of the THIA derivatives with their corresponding IC₅₀ values. Including a table summarizing this data in the manuscript  as one table would greatly help readers understand the correlation.
  10. Figures 11, 12, and 13 are difficult to interpret and of poor quality. Please replace them with clearer, higher-quality images that effectively convey the intended information. Also provide the covalent docking score.

Author Response

1. Compare the known cysteine protease inhibitors with THIA derivatives as controls in the inhibition studies. Positive and negative controls are missing or not described

Response: On page 8, we introduced a paragraph comparing the classical irreversible inhibitor of cysteine proteases, earlier reported [Barret et al., 1982], with the THIA compounds. The negative effect of THIA compounds on bromelain was also discussed, which was compared with E-64 on the same enzyme.

2. The papain Ki​ and IC₅₀ data presented in the manuscript are difficult to interpret. The relationship between binding affinity (Ki) and inhibition concentration (IC₅₀) is not clearly explained or correlated. Please clarify this distinction and provide context to help readers understand the data.

Response: We provided the conceptualizations of Ki and IC50 in the legend to Table S2.

3. Inhibitor studies are performed at Different pH, papain in sodium phosphate (pH 6.0) Cathepsin in Sodium acetate (pH 5.0), 3Clpro in Tris (7.5), why given the unstability of THIA derivatives?

Response These were the optimal conditions for each enzyme, and the substrates were stable, since we did not observe fluorescence variations during all the pre-incubation steps of our studies.

4. Provide a good resolution image for Figure S4 and mention the number of replicates.

Response: We improved this figure by indicating errors and replication numbers.

5. The degradation studies of THIA derivatives are interesting and well described; however, the reason for the rapid degradation of these compounds is not explained. Is the degradation primarily due to pH effects, hydrolysis, or another mechanism? Please clarify why THIA-4 and THIA-9 exhibit complete degradation within 24 hours in DMSO. Additionally, THIA-1 and THIA-8 display three peaks at 12 hours with retention times between 7.5 and 13 minutes—please explain the origin of these peaks in the manuscript. For clarity, include a table summarizing the percentage degradation of each derivative over time.

Response: The decomposition of THIA compounds in or by DMSO is, in fact, of interest; however, this subject is out of the scope of our paper for a detailed study (as already mentioned in the main text). A thorough analysis of the chemistry of these reactions is in progress. Then, we limited ourselves here only to presenting the conditions of stability.

6. The manuscript comments on the effect of R and R₂ substituents on the potency and selectivity of THIA derivatives; however, only a limited number of compounds were investigated, which restricts the strength of the conclusions that can be drawn. Given the computation resources and availability, it would be valuable to design and perform molecular docking studies on a broader range of THIA derivatives.

Response: We agree with the reviewer that it would be valuable to design new THIA derivatives on the bases of the computational models developed by the present study. However, the newly designed molecules will be the object of future paper(s) also describing the synthesis and biological evaluation of the newly designed compounds. A new paragraph entitled Future Directions has been added on page 33, where strategies for future modifications are described to better guide subsequent research.

7             Mention the binding velocity for Z-phe Arg-MCA, Abz-SAVLQSGFRK(Dnp)NH2 substrates with respective enzymes? how stable are these peptides in assay conditions? And selectivity towards other proteases.

Response: In the legend of Table 2, we introduced the kinetic parameters of these substrates by the respective enzymes. Regarding stability, the fluorescence Abz-SAVLQSGFRK(Dnp)NH2  and Z-Phe-Arg-MCA solutions remained unchanged under control experiments and in stock solutions. Abz-SAVLQSGFRK(Dnp)NH2 with cleavage at Q-S bond is highly specific for 3Clpro, and the best sequence for 3Clpro as we demonstrated by the FRET peptide library data in Table 3. In contrast, Z-Phe-Arg-MCA is typically used as an operational fluorescent substrate.

8. In the methods section, preactivation of the cysteine protease (CP) with DTT is described. Please specify the concentration of DTT used and the incubation time for this step. Additionally, considering that DTT requires extra steps such as removal by gel filtration to prevent interference in downstream assays, why was TCEP not used as an alternative reducing agent?

Response: We used 5 mM of DTT for 30 min and then proceeded to gel-filtration (added in the Results section on page 4). In fact TCEP is an alternative as a reducing agent. However, we preferred not to introduce an extra element in the kinetic experiments.

9. Compare and discuss the docking scores of the THIA derivatives with their corresponding IC₅₀ values. Including a table summarizing this data in the manuscript  as one table would greatly help readers understand the correlation.

Response: According to the reviewer’s suggestion, Table 4 has been modified by adding a column reporting the IC50 values against 3CLpro and docking results  are discussed   in relation to their IC50.

10. Figures 11, 12, and 13 are difficult to interpret and of poor quality. Please replace them with clearer, higher-quality images that effectively convey the intended information. Also provide the covalent docking score.

Response: Figures 11, 12, and 13 (in the revised text figures 14-16) were modified, and we discussed the data of them in more detail.

Reviewer 3 Report

Comments and Suggestions for Authors

The manuscript titled 1,2,4-Thiadiazolidin-3,5-diones as inhibitors of cysteine proteases deals with  inhibitory activity of 1,2,4-thiadiazolidin-3,5-diones against a panel of cysteine proteases. The manuscript is well organized and research methodology is adequately applied. However, before publication the authors must resolve following issues:

  1. To be able to comment structure/biological activity relationship, the authors must include more examples of 1,2,4-thiadiazolidin-3,5-diones into the manuscript. If compounds used in this study are synthetically obtained, the reference related to the adequate synthetic protocol should be added.
  2. In Table 1, the hyphen before the substituent OCH3 (compound THIA-8) should be removed
  3. The quality of Scheme 1 must be improved.
  4. A paragraphs describing future directions of this research should be added at the end of manuscript – for example, cell-based antiviral activity assays, wider panel of human cysteine proteases planed for testing, etc.
  5. Especially for SARS-CoV-2 protease, it is important to test whether THIA is time-dependent and distinguish between reversible vs. irreversible inhibition.
  6. Molecular modeling can be extended by molecular dynamics simulation in order to validate docking results in dynamic environment.

Author Response

1. To be able to comment structure/biological activity relationship, the authors must include more examples of 1,2,4-thiadiazolidin-3,5-diones into the manuscript. If compounds used in this study are synthetically obtained, the reference related to the adequate synthetic protocol should be added.

Response: The compounds investigated in this study are synthetically obtained, and the appropriate reference to the original synthetic protocol has been included in the manuscript (reference 3). In addition, a brief description of the synthetic procedure, together with the modified purification process applied to compound THIA-3, is reported in the Experimental section. New compounds, designed on the bases of the results of this study, will be published in future manuscripts.

2. In Table 1, the hyphen before the substituent OCH3 (compound THIA-8) should be removed

Response: The hyphen preceding the OCH₃ substituent in compound THIA-8 has been removed.

3. The quality of Scheme 1 must be improved.

Response: Scheme 1 was edited and changed for clarity.

4. A paragraph describing future directions of this research should be added at the end of manuscript – for example, cell-based antiviral activity assays, wider panel of human cysteine proteases planned for testing, etc.

Response:: We introduced at the end of the manuscript a new section: “5. Future directions”.

5. Especially for SARS-CoV-2 protease, it is important to test whether THIA is time-dependent and distinguish between reversible vs. irreversible inhibition.

Response: We introduced the time-dependent inhibition of 3Clpro data in Figure S3

6. Molecular modeling can be extended by molecular dynamics simulation in order to validate docking results in dynamic environment.

Response: According to the reviewer’s suggestion, in order to validate the docking results in dynamic environment, the selected docked complexes were subjected to Steered Molecular Dynamics (SMD) simulations. The results obtained from SMD simulations have been reported in a new paragraph of the “Results and Discussion” section, entitled 2.5.3. Steered Molecular Dynamics (pages 18–20) and presented in two additional figures introduced in the main text, and in five new figures added in the SI. The experimental procedure has been added to the “Materials and Methods” section (pages 30–31).

Round 2

Reviewer 3 Report

Comments and Suggestions for Authors

I believe that after all changes have been accepted and implemented by authors, this manuscript could be published in this reputable journal.